# Magnitude and correlates of alcohol use disorder in south Gondar zone, northwest Ethiopia: A community based cross-sectional study

Getasew Legas[1]*, Sintayehu Asnakew[1], Amsalu Belete[1], Getnet Mihretie Beyene[1], Gashaw Mehiret Wubet[2], Wubet Alebachew Bayih[3], Ermias Sisay Chanie[3], Agimasie Tigabu[3], Tigabu Dessie[3]

1 Department of Psychiatry, School of Medicine, College of Health Sciences, Debre Tabor University, Debre Tabor, Ethiopia, 2 Department of Medicine, School of Medicine, College of Health Sciences, Debre Tabor University, Debre Tabor, Ethiopia, 3 Department of Nursing, College of Health Sciences, Debre Tabor University, Debre Tabor, Ethiopia

* getasewlegas@gmail.com

## Abstract

### Background

Alcohol use disorder is the major public health problem in low- and middle-income countries that account for up to 70% of alcohol related premature mortality in the region. Therefore, the aim of this study was to assess the magnitude of alcohol use disorder and its associated factors among adult residents in south Gondar zone, Northwest Ethiopia.

### Methods

A community-based cross-sectional study was conducted among 848 adult residents of the south Gondar zone from January 13 to February 13, 2020. A multistage sampling technique was used to recruit study participants. We assessed alcohol use disorder (AUD) using the alcohol use disorder identification test (AUDIT). A binary logistic regression model was employed to identify factors associated with AUD.

### Results

The prevalence of alcohol use disorder over the last 12-months was found to be 23.7% (95% CI: 20.9, 26.7). Being male (AOR = 4.34, 95 CI: 2.800, 6.743), poor social support (AOR = 1.95, 95 CI: 1.098, 3.495), social phobia (AOR = 1.69, 95 CI; 1.117, 2.582), perceived high level of stress (AOR = 2.85, 95 CI; 1.829, 34.469), current cigarette smoking (AOR = 3.06, 95 CI; 1.764, 5.307) and comorbid depression (AOR = 1.81, 95 CI; 1.184, 2.778) were significantly associated with alcohol use disorder.

**Data Availability Statement:** All relevant data are within the paper.

**Funding:** The authors received no specific funding for this work.

**Competing interests:** The authors have declared that no competing interests exit.

**Abbreviations:** AUDIT, alcohol use disorder identification test; DALYs, Disability Adjusted Life Years; PHQ, Patient Health Questionnaire; SPI, Social phobia inventory scale; OSS-3, Oslo-3 Social Support Scale; AUD, alcohol use disorder; PSS, perceived stress scale.

## Conclusion

The prevalence of alcohol use disorder is high among adult residents of the south Gondar zone and associated with many factors. So, it needs public health attention to decrease the magnitude of alcohol use disorder in Ethiopia.

## Background

Alcohol use disorder is defined as a persistent pattern of alcohol use characterized by taking a large amount of alcohol drinking, unsuccessful effort to cut down, strong desire to use, increased amount of drinking to achieve the desired effect, and experience of withdrawal symptoms after reduction of drinking occurring within a 12-month period [1].

Alcohol is the most prevalent substance use disorder, up to 99.2 million disability-adjusted life years (DALYs) were attributed to alcohol use disorder [2]. Approximately more than 2 billion adults (48% of the adult population) use alcohol [3, 4]. Globally, 4% of the global burden of disease was contributed by alcohol consumption, which is equivalent to tobacco smoking, but alcohol is more contributes to premature death in young adults than tobacco smoking [5].

In recent years, high-income countries have increased their attention to alcohol-related harm by implementing governmental regulations to reduce alcohol consumption [6, 7]. It contributes to more than 200 alcohol-related preventable diseases. It was also responsible for 7.6% and 4% of alcohol-related deaths in men and women respectively [8, 9]. However, up to 70% of alcohol-related premature mortality occurs in low- and middle-income countries [10, 11].

In sub-Saharan Africa, alcohol consumption was higher than the global consumption rate (7.4% vs 6.2%) and the consumption rate was 42% higher than the global rate per adult drinker [12].

The magnitude of alcohol use disorder is different across the world. In the United States, approximately about 13.9% and 29.1% of persons aged 18 years and older suffer from 12-month and lifetime AUD respectively [13]. A study conducted in rural Brazil reported that 18.4% of subjects had AUD [14]. Another data in Brazil showed, almost half (50%) of participants had AUD [15]. A community-based study done in India showed that the magnitude of alcohol use disorder was 9.4% [16]. In Nepal, a number of researches were conducted in the general population and the magnitude of AUD ranged from 7.3% to 25.8% [16–18].

In Africa, two studies were done in Uganda and Nigeria, the magnitude of alcohol use disorder was 9.8% and 39.5% respectively [19, 20]. A number of factors were found to affect alcohol use disorder. Different kinds of the literature showed that male sex, illiteracy, low level of education, perceived high level of stress, poor social support, stressful life events, cigarette smoking, comorbid depression, and social phobia were significantly associated with alcohol use disorder [21–30].

In Ethiopia, the magnitude of alcohol use disorder in the general population ranges from 12.4% to 21% [31–33]. So, assessing the magnitude of alcohol use disorder at the community level had a great significance to enforce the government and stakeholders to implement mental health services in primary health care settings to manage alcohol use disorder. So, this study is designed to determine the magnitude and factors associated with alcohol use disorder among adult residents of the south Gondar zone, northwest Ethiopia.

## Methods and materials

### Study settings and populations

A community-based cross-sectional study was conducted from January 13 to February 13, 2020. The study was conducted in the South Gondar zone, northwest Ethiopia. South Gondar zone was divided into fifteen districts/woredas with an estimated population of 2,051,738. From those, 1,041,061 were men and 1,010,677 were women. Debre Tabor town is the capital city of the south Gondar zone. The town is 100 km far from Bahir Dar (the capital city of the Amhara region) and 667 km far from the capital city of Ethiopia. Currently, the south Gondar zone had one referral hospital, seven district hospitals, and 94 health centers but mental health service (outpatient and inpatient treatment) is delivered by only three governmental hospitals.

### Study participants

The study was conducted among adult residents whose age was 18 years and above in the south Gondar zone, northwest Ethiopia. Individuals who were seriously ill and unable to communicate were excluded from the study. We determined the sample size by using the single population proportion formula assuming that 21% of adult residents in Ethiopia might have alcohol use disorder at 95% CI, 4% margin of error, and adding a 10% non-response rate. Considering of design effect of 2, the final sample size was 875.

### Sampling

A multistage sampling technique was used to select 848 study participants. In a total of 15 districts/woredas, we selected three districts by simple random sampling technique. Then, we selected three sub-districts/kebeles in each of the selected districts of the south Gondar zone. In each of the selected sub-districts, we selected 875 households proportionally. In the case of more than one individual in a household, select one of them by lottery method. The number of households was obtained from health extension workers (**Fig 1**).

### Measurements

Sociodemographic factors age, sex, marital status (never married, married and living together, married and not living together, divorced, or widowed), living circumstance (living with family or alone), educational status (unable to read and write, primary education, middle school or college and above), residence (rural or urban), and occupational status (employed or unemployed) were adopted from different literatures conducted in Ethiopia.

To identify AUD, we used a 10-item alcohol use disorder identification test questionnaire (AUDIT). Each item of AUDIT was rated on a five-point scale, ranges from 0 to 40. A total score of 1–7 considered social drinker, the total of score 8 or more indicates probable alcohol use disorder, the total score of 8–15 indicates hazardous alcohol use, a score of 16–19 indicates harmful alcohol use and a score of 20 or more indicates probable alcohol dependence in the last 12-months [34]. The presence of comorbid depression in the last two weeks was assessed by the patient health questionnaire (PHQ-9). The tool has nine items and each item has rated on a four-point scale, 0 (not at all), 1 (several days), 2 (more than half the days), and 3 (nearly every day) with the total score ranging from zero to 27. A score of five or more on the PHQ-9 questionnaire considered as having comorbid depression [35]. Individual-level of stress in the last one month was measured using 10 items of perceived stress scale questionnaire (PSS). Each item of PSS has rated on a five-point scale, 0(never), 1(almost never), 2 (sometimes), 3 (fairly often), 4 (very often) with a total score ranging from zero to 40. Scoring of 0–13 considered a low level of stress, scoring of 14–26 considered a medium level of stress, and scoring of

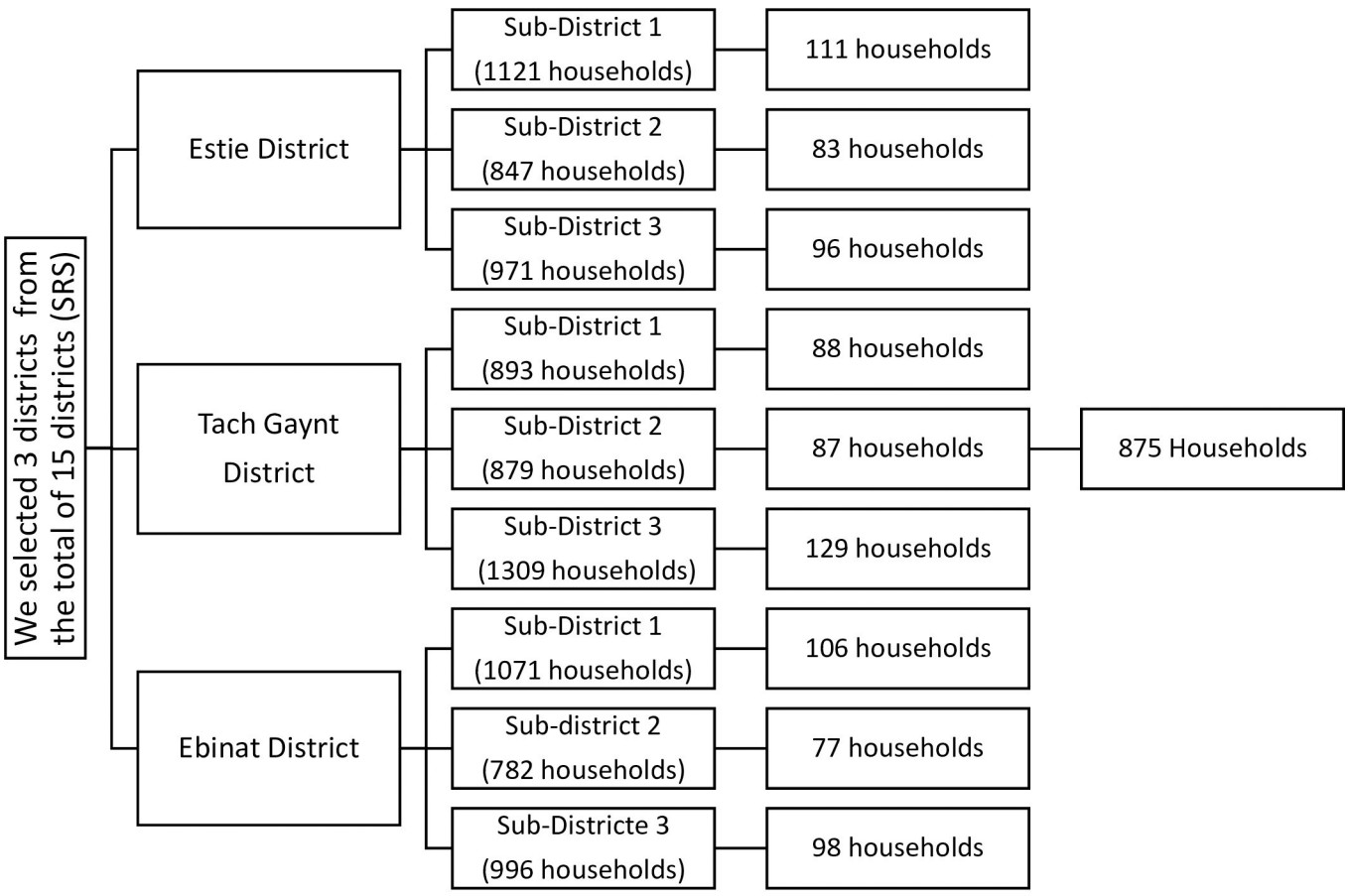

**Fig 1. Schematic presentation of sampling procedure in adult residents of south Gondar zone, northwest Ethiopia, 2020.** n = 875.

27–40 considered a high level of stress. Previous studies carried out based on this cut-off point [36–39]. The level of social support was measured using a three-item Oslo social support scale which has three items with a range of between three and fourteen." scoring of 12–14 = Strong support" score of 9–11 = Moderate social support and", a score of "3–8" Poor social support scale [40]. Social phobia was assessed by a 17-item social phobia inventory scale (SPIN). SPIN has 17- items rates from 0 (not at all) to 4 (extremely) with a total score ranges from 0 to 68. A score of 21 or more is considered as having social anxiety disorder [41]. The tool was validated both for adults and adolescents in different countries [42, 43].

## Data collection

The data was collected by nine psychiatry nurses after three days of training and supervised by three-degree holder psychiatry professionals. The interviewer administered questionnaire was used after translated into Amharic language (local working language).

## Data processing and analysis

The collected data were checked for completeness and consistency and entered into Epi-data V.3.1. Then, exported to SPSS window V.21 for analysis. Descriptive, bivariate, and multivariate logistic regression analysis was used to identify factors associated with the outcome variable. A p-value of $< 0.05$ at 95% CI with adjusted OR is considered as statistically significant.

### Ethical consideration

Ethical clearance was obtained from the ethical review committee of Debre Tabor University. Confidentiality was maintained by omitting the name and address of the participants. Written informed consent was taken after explaining the purpose of the study. A formal written permission letter was obtained from the south Gondar zone health department.

## Result

### Sociodemographic characteristics

A total of 848 participants were interviewed with a response rate of 96.91%. The majority of the respondents, 528(62.3%) were male. 824 (97.2%) of participants were Amhara by ethnicity. Regarding religion, 804 (97.2%) were orthodox Christian followers. 320 (37.7%) were married and living together. 600 (70.8%) were employed. Most of the respondents 520 (61.3%) were from urban areas. 280(33%) were attended primary education and 616(72.6%) were living with family (**Table 1**).

### Clinical, psychosocial, and substance use characteristics

Of the total 848 respondents, 312 (36.8%) had a family history of alcohol use, and the majority, 376(44.3%) had poor social support. Two hundred twenty-four (26.4%) had social phobia and

**Table 1. Socio-demographic characteristics of the participants in south Gondar zone, northwest Ethiopia, 2020 (n = 848).**

| Variable | Category | Frequency | Percentage |
|---|---|---|---|
| Age | 18–24 | 208 | 24.5% |
| | 25–34 | 200 | 23.6% |
| | 35–44 | 208 | 24.5% |
| | 45–44 | 104 | 12.3% |
| | > = 55 | 128 | 15.1% |
| Sex | Male | 528 | 62.3% |
| | Female | 320 | 37.7% |
| Ethnicity | Amhara | 824 | 97.2% |
| | other | 24 | 2.8% |
| Educational status | Unable to read and write | 216 | 25.5% |
| | 1–8 grade | 176 | 20.8% |
| | 9–12 grade | 280 | 33% |
| | Diploma & above | 176 | 20.8% |
| Religion | Orthodox Christian | 804 | 94.8% |
| | Other | 44 | 5.2% |
| Marital status | Never married | 304 | 35.8% |
| | Married & living together | 320 | 37.7% |
| | Married & not living together | 56 | 6.6% |
| | Divorced | 112 | 13.2% |
| | Widowed | 56 | 6.6% |
| Living circumstance | With family | 616 | 72.6% |
| | Alone | 232 | 27.4% |
| Occupational status | Employed | 600 | 70.8% |
| | Non-employed | 248 | 29.2% |
| Residence | Rural | 328 | 38.7% |
| | Urban | 520 | 61.3% |

175(20.6%) having co-morbid depression. Regarding substance use, 322(38.4%) of respondents experienced Khat (leaves) chewing in their lifetime and one hundred eight (12.7%) were smoke cigarettes in the last three months **(Table 2)**.

## Magnitude of alcohol use disorder

The magnitude of alcohol use disorder in adult residents of the south Gondar zone was 23.7% (95% CI: 20.9, 26.7). From these, 140 (16.50%) had hazardous alcohol use, 44 (5.2%) had harmful alcohol use, and 17 (2%) had probable alcohol dependence (**Fig 2**).

## Factors associated with alcohol use disorder

To determine the association of independent variables with alcohol use disorder. Bivariate and multivariable logistic analysis was carried out. Poor social support, living alone, having co-morbid depression, social phobia, male sex, current use of khat, current cigarette smoking, current use of cannabis, and high level of stress was associated with alcohol use disorder on bivariate analysis. Poor social support, social phobia, co-morbid depression, high level of stress, male sex, and current cigarette smoking were associated with alcohol use disorder on multivariable analysis.

This study showed that alcohol use disorder was 1.81 times higher among respondents who had depression compared with those who had no depression (AOR = 1.81, 95 CI; 1.184, 2.778).

**Table 2. Clinical, psychosocial, and substance use characteristics of the participants in south Gondar zone, northwest Ethiopia, 2020 (n = 848).**

| Variables | Category | Frequency | Percent % |
|---|---|---|---|
| Family history of mental illness | Yes | 87 | 10.3% |
| | No | 761 | 89.7% |
| Family history of alcohol use | Yes | 312 | 36.8% |
| | No | 536 | 63.2% |
| Ever use of Khat | Yes | 322 | 38.4% |
| | No | 526 | 61.6% |
| Ever use of cigarette smoking | Yes | 214 | 25.2% |
| | No | 634 | 74.8% |
| Ever use of cannabis | Yes | 153 | 18% |
| | No | 695 | 82% |
| Current use of Khat | Yes | 188 | 22.2% |
| | No | 660 | 77.8% |
| Current use of cigarette smoking | Yes | 108 | 12.7% |
| | No | 740 | 87.3% |
| Current use of cannabis | Yes | 32 | 96.2% |
| | No | 816 | 3.8% |
| Depression | Yes | 175 | 20.6% |
| | No | 673 | 79.4% |
| Social phobia | Yes | 224 | 26.4% |
| | No | 624 | 73.6% |
| Social support | Poor | 376 | 44.3% |
| | Moderate | 312 | 36.8% |
| | Strong | 160 | 18.9% |
| Individual level of stress | Low | 640 | 75.5% |
| | Moderate | 160 | 18.9% |
| | High | 48 | 5.7% |

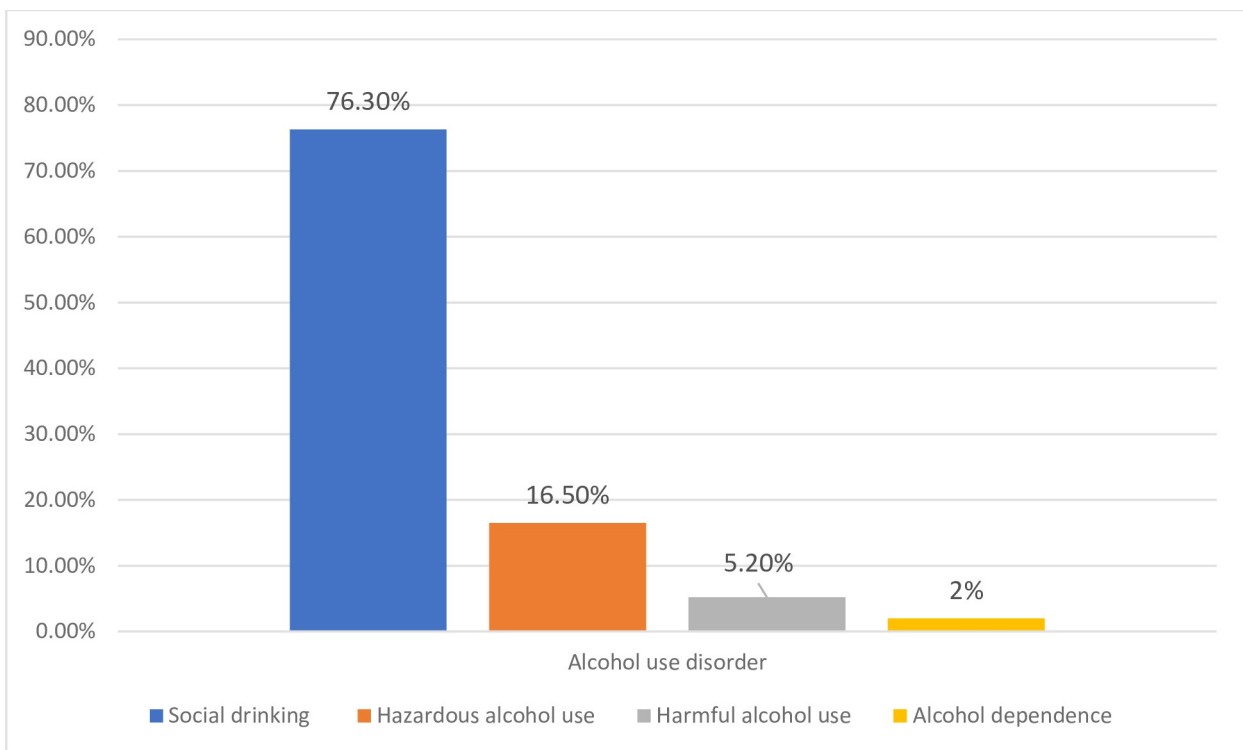

**Fig 2. Bar chart showing that the level of alcohol use in adult residents of south Gondar zone, northwest Ethiopia, 2020 (N = 848).**

Alcohol use disorder was more common among respondents who had poor social support compared with the respondents who had strong social support (AOR = 1.95, 95 CI: 1.098, 3.495). Regarding the psychosocial factors, alcohol use disorder was 2.85 times higher among respondents who had a high level of stress compared with those who had a low level of stress (AOR = 2.85, 95 CI; 1.829, 4.469). Social phobia was the other factor that was found to be significantly associated with alcohol use disorder (AOR = 1.69, 95 CI; 1.117, 2.582). Current smoking cigarette was also a major factor in the development of alcohol use disorder (AOR = 3.06, 95 CI; 1.764, 5.307). Our study also identified male gender had a statistically significant positive correlation with alcohol use disorder (AOR = 4.34, 95 CI; 2.800, 6.743) (**Table 3**).

## Discussion

The magnitude of alcohol use disorder in adult residents of the south Gondar zone was 23.7% (95% CI: 20.9, 26.7). The finding of this study was in line with a study carried out in Ethiopia, 21% [33], Dharan town, 25.8% [16], and Chitwan District of Nepal 23.8% [18]. However, the finding of the current study was lower than a study conducted in Brazil 50% [15], and Nigeria 39.5% [19].

Conversely, the finding of this study was higher than the findings in the USA 13.9% [13], India, 9.4% [44], Uganda 9.8% [20], Colombia 9% [45], and studies done in other parts of Ethiopia, (12.3%, in southwest Ethiopia and 13.9%, in southern rural, Ethiopia) [31, 32]. One of the possible explanations for the difference in the previous studies carried out in Ethiopia, the majority of the study participants in Jimma town was Muslim and Protestant, compared with the current study sample which was over 97% of the study sample was orthodox Christian followers. In Sodo, Gurage Zone, South Ethiopia, more than half of the sample was females

**Table 3. Factors associated with Alcohol use disorder among residents of south Gondar zone, northwest Ethiopia, 2020 (N = 848).**

| Variables | Category | Alcohol use disorder | | COR 95% CI | AOR 95% CI | P-value |
|---|---|---|---|---|---|---|
| | | Yes | No | | | |
| Sex | Male | 168 | 360 | 4.05(2.709–6.080) | 4.34(2.80–6.743) * | **0.000** |
| | Female | 33 | 287 | 1 | 1 | |
| Perceived stress | Low | 114 | 526 | 1 | 1 | |
| | Moderate | 71 | 89 | 2.30(1.225–4.346) | 1.76(0.849–3.655) | 0.128 |
| | High | 16 | 32 | 3.68(2.538–5.339) | 2.85(1.829–4.469) * | **0.008** |
| Social phobia | Yes | 91 | 133 | 3.19(2.283–4.478) | 1.69(1.117–2.582) * | **0.013** |
| | No | 110 | 514 | 1 | 1 | |
| Living circumstance | With family | 124 | 492 | 1 | 1 | |
| | Alone | 77 | 155 | 1.97(1.407–2.761) | 1.23(0.811–1.871) | 0.328 |
| Depression | Yes | 70 | 105 | 2.75(1.929–3.944) | 1.81(1.184–2.778) * | **0.006** |
| | No | 131 | 542 | 1 | 1 | |
| Social support | Poor | 102 | 274 | 2.33(1.410–3.866) | 1.95(1.098–3.495) * | **0.023** |
| | Moderate | 77 | 235 | 2.05(1.224–3.451) | 1.50(0.851–2.671) | 0.159 |
| | Strong | 22 | 138 | 1 | 1 | |
| Current cigarette smoking | Yes | 49 | 59 | 3.21(2.114–4.883) | 3.06(1.764–5.307) * | **0.000** |
| | No | 152 | 588 | 1 | 1 | |
| Current use of khat | Yes | 59 | 129 | 1.66(1.164–2.391) | 0.93(0.606–1.426) | 0.740 |
| | No | 142 | 518 | 1 | 1 | |
| Current use of cannabis | Yes | 13 | 19 | 2.28(1.108–4.715) | 0.65(0.249–1.737) | 0.397 |
| | No | 188 | 628 | 1 | 1 | |

1 (reference group),

* (p<0.05), COR (crude odds ratio), AOR (adjusted odds ratio)

compared to the current study sample, only 37.7% of the study sample was females. However, females are less likely to involve in public place drinking and heavy drinking due to cultural influence. As a result, females are less likely to have alcohol use disorder. The other possible reasons for this discrepancy might be the difference in the number of participants, differences in the assessment tool used, economic differences, and cultural differences that might have a higher contribution to the discrepancy.

We found an association between social phobia and alcohol use disorder (AOR = 1.69, 95 CI; 1.117, 2.582). The result is similar to the findings of studies conducted in the USA [25, 27, 46]. Subjects who had social anxiety disorder could be drink alcohol to control their fears and anxiety [47].

In the current study poor social support was predictive of alcohol use disorder (AOR = 1.95, 95 CI: 1.098, 3.495). The finding of this study was in line with studies carried out in Sweden [28], and previous Ethiopian studies [22, 29, 30]. The possible reason might be due to lack of experience in social relationships, social and psychological support from their neighborhood and relatives can lead to using alcohol.

Our study also identified male gender had a statistically significant positive correlation with alcohol use disorder (AOR = 4.34, 95 CI; 2.800, 6.743). This finding is supported by studies done in India [44], Brazil [14, 15], and other parts of Ethiopia [22, 32]. Heavy drinking is mostly occurred in men compared with women and biological differences among the two genders might be contributing to the discrepancy. Not only biological factors, but females are also less likely to involve in public places drinking due to cultural influence. So, it could be less likely to have alcohol use disorder [19, 48].

In our study, alcohol use disorder was found to be significantly associated with comorbid depression (AOR = 1.81, 95 CI; 1.184, 2.778). The study is also in line with studies done in Nepal [18], Lebanon [23], India [26], Greek [24], and Uganda [49]. The possible reason might be subjects who had depressive symptoms might be using alcohol to relieve their symptoms [50, 51].

This study showed that alcohol use disorder was 2.85 times higher among respondents who had a high level of stress compared with those who had low level of stress (AOR = 2.85, 95 CI; 1.829, 4.469). The current finding was supported by a study carried out in the UK [52], Korea [53], Greek [24], Lebanon [23], Having someone high level of stress can lead to drinking as a coping mechanism of stress [53].

Finally, we found also an association between current cigarette smoking and alcohol use disorder (AOR = 3.06, 95 CI; 1.764, 5.307). The finding of this study was similar to studies conducted in Nepal [18], Brazil [14], Nigeria [19], India [26], Greek [24], and other parts of Ethiopia [22, 30, 32, 54]. The possible reason might be the two substances share a rewarding effect in the activation of the mesolimbic pathway [55].

## Limitations of the study

This study was a cross-sectional study we tried to assess different factors that may predict alcohol use disorder. However, the temporal relationship cannot be concluded by this study design. In addition, recall bias might be also the other limitation of this study. Although the alcohol use disorder identification test (AUDIT) was not validated in the general population of Ethiopia. Despite these limitations, this study had strengths, including many factors that were not addressed in previous Ethiopian studies.

## Conclusion

Alcohol use disorder in south Gondar zone adult residents was found to be high. Comorbid depression, male gender, perceived high level of stress, poor social support, social phobia, and cigarette smoking were significantly associated with alcohol use disorder. Therefore, the researchers recommend regular screening of alcohol use disorder by trained health professionals at the community level and the referral linkage with mental health services should be strengthened.

## Supporting information

**S1 File. Data collection tool to assess magnitude and corelates of alcohol use disorder in south Gondar zone, northwest Ethiopia.**
(DOCX)

## Acknowledgments

We would like to acknowledge data collectors and study participants. We would like also to thank Debre Tabor University for ethical clearance.

## Author Contributions

**Conceptualization:** Sintayehu Asnakew.

**Formal analysis:** Getasew Legas.

**Methodology:** Getasew Legas.

**Writing – original draft:** Getasew Legas, Gashaw Mehiret Wubet, Wubet Alebachew Bayih, Ermias Sisay Chanie, Tigabu Dessie.

**Writing – review & editing:** Amsalu Belete, Getnet Mihretie Beyene, Agimasie Tigabu.

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
