## [Decision Letter · Decision Letter 0]

17 Jun 2021

PONE-D-21-07650

Magnitude and correlates of alcohol use disorder in south Gondar zone, northwest Ethiopia: a community based cross-sectional study

PLOS ONE

Dear Dr. Legas,

Thank you for submitting your manuscript to PLOS ONE. After careful consideration, we feel that it has merit but does not fully meet PLOS ONE’s publication criteria as it currently stands. Therefore, we invite you to submit a revised version of the manuscript that addresses the points raised during the review process.

We look forward to receiving your revised manuscript.

Kind regards,

Markos Tesfaye, M.D., Ph.D

Academic Editor

PLOS ONE

Additional Editor Comments (if provided):

The report presents results from a community survey of alcohol use disorder using screening tool AUDIT through interviews of 848 residents in north west Ethiopia. The authors also presented logistic regression analysis to identify factors associated with alcohol use disorder. The study attempts to address an important public health issue globally as well as in Africa. While the work and results may be relevant to wide range of audience, some of the methods and data interpretation is flawed. Therefore, the authors need to consider the following issues to improve the scientific value of the report.

1. The manuscript has many typographical and grammatical errors that interfere with comprehension. The manuscript needs extensive language revision preferably by a native English speaker.

2. The methods section needs to provide more detailed information about the psychometric properties of AUDIT rather than qualitative statement such as "... is the most preferable tool to identify individuals with AUD"

3. The methods section refers to PHQ-9 validation with a suggested cut-off of 5. The reference actually recommended cut-off of 10 and above. Therefore, the data needs to be re-analyzed using an appropriate cut-off.

4. Similarly, the citation made for PSS-10 does not provide any of the cut-offs mentioned in the manuscript. You may provide correct reference or re-analyze the data using stress scores as continuous variable.

5. It is not clear why the interviews were used rather than self-administered data collection. In addition, what was the setting of data collection? What is the potential effect of using health professionals as interviewers of this sensitive issue? The limitation need to address any biases arising from this.

6. The ethics statement need to be included in the manuscript submission system rather than just "human participants". In addition, information on what was done for any participant with suicide risk and alcohol dependence is important ethical issue. Please clarify what was done or why no action was taken.

7. In the abstract, results, and discussion, the authors use terms suggestive of causation which cannot be inferred from this cross-sectional survey. Terms such as 'risk factor', 'predictor', 'odds of developing', 'likely to develop', etc. need to be replaced with 'associated with' or other appropriate terms.

8. The discussion needs to provide in depth interpretation and implications than simple comparisons of prevalence. You may consider to provide if there are local factors known to increase alcohol use or indicators of complications such as road traffic accident data, etc.

Minor comments:

One or two decimal places are enough for the odds ratios.

Journal Requirements:

 Thank you for stating the following financial disclosure:

No.

b) State what role the funders took in the study. If the funders had no role in your study, please state: “The funders had no role in study design, data collection and analysis, decision to publish, or preparation of the manuscript.

d) If you did not receive any funding for this study, please state: The authors received no specific funding for this work.

5. Please amend the manuscript submission data (via Edit Submission) to include author Sintayehu Asnakew, Amsalu Belete, Getnet Mihretie, Gashaw Mehiret, Wubet Alebachew, Ermias Sisay, Agimasie Tigabu, Tigabu Dessie.

Reviewers' comments:

Reviewer's Responses to Questions

**Comments to the Author**

1. Is the manuscript technically sound, and do the data support the conclusions?

Reviewer #1: Yes

Reviewer #2: Yes

2. Has the statistical analysis been performed appropriately and rigorously? 

Reviewer #1: Yes

Reviewer #2: Yes

3. Have the authors made all data underlying the findings in their manuscript fully available?

Reviewer #1: Yes

Reviewer #2: Yes

4. Is the manuscript presented in an intelligible fashion and written in standard English?

Reviewer #1: Yes

Reviewer #2: Yes

5. Review Comments to the Author

Reviewer #1: Thanks for your invitation to review this study. Since evidences related to substance use disorders like alcohol are scared in low and middle-income countries like ours particularly in Amhara region. Thus, this study added a valuable contribution to the medical field. So, to make this useful for the general public the authors need to incorporate the comments mentioned below

Language

I advise extensive editing/proof-reading of the written English and re-drafting before the paper can be considered suitable for publication, b/se there are many odd phares and language errors.

Introduction

1. You put repeated sentence at your introduction and citing different references (reference number 6&7). For example, in recent years, high-income countries have increased their attention to alcohol-related harm by implementing governmental regulations to reduce alcohol consumption. In recent years, high-income countries have increased their attention to alcohol-related harm by implementing governmental regulations to reduce alcohol consumption.

Methods and materials

1. Better to say "study setting" rather than study settings

Study participants

1. Have you included participants with mental illness?

2. Under study setting and populations, it is good to mention the numbers and the type of health institutions in South Gondar Zone. In addition to this I strongly recommend you to include the type of mental health services delivered in those hospitals

Measurements

1. Why you preferred to measure alcohol used disorder by using CAGE other than AUDIT? More clarification is required.

Data collection

2. How study participants were reached e.g., door to door knocking/ or telephone interview?

Discussion

3. In paragraph 1 and 2 of your discussion were mentions Brazil twice in contradicting sentences " However, the finding of the current study was lower than a study conducted in Brazil 50% (15), and Conversely, the finding of this study was higher than the findings in USA 13.9% (13), India, 9.4% (41), Uganda 9.8% (20), Colombia 9% (42), Brazil, 18.4% (14) ...

4. On paragraph 4 of your discussion, References are particularly useful to substantiate statements which could be considered strengthen your justification e.g. "The possible reason might be due to lack of experience in social relationship, social and psychological support from their neighborhood and relatives can lead to use alcohol"

5. How did you manage the participant who are eligible for dependence treatment during the data collection time?

Reviewer #2: The statement of the problem at the last end of the of the background is not convincing because the community is part of the general population.

On the sampling, the authority used in using (21%) in calculating the size has to be indicated in the manuscript.

The authors used sociodemographic data in the analysis, but have not mentioned the instrument used to collect that demographic data under measurements.

Under results on sociodemographic, the author use figures to start sentence, they need to write those figures in words.

Under limitation, the authors failed to indicate the strength of their research and the application of their findings to solving the alcohol disorder in the study district.

6. PLOS authors have the option to publish the peer review history of their article (what does this mean?). If published, this will include your full peer review and any attached files.

Reviewer #1: **Yes: **Mengesha Birkie

Reviewer #2: No

---

## [Author Response · Author response to Decision Letter 0]

28 Jul 2021

Response for the editor’s and reviewers’ enquiry 

Dear Editor,

Thank you for your valuable comments and recommendations. 

If further correction is needed from the editor, we are ready to do the requested enquires again including re-analysis before the decision has made. 

1/ For the Editor

1. The manuscript has many typographical and grammatical errors that interfere with comprehension. The manuscript needs extensive language revision preferably by a native English speaker.

 Great thanks! We tried to modify the language throughout the paper.

2. The methods section needs to provide more detailed information about the psychometric properties of AUDIT rather than qualitative statement such as "... is the most preferable tool to identify individuals with AUD" 

 Thank you for your constructive comment. Modified based on the comment. Please see page 6.

3. The methods section refers to PHQ-9 validation with a suggested cut-off of 5. The reference actually recommended cut-off of 10 and above. Therefore, the data needs to be re-analyzed using an appropriate cut-off.

 Thank you dear editor! Yes, it is validated at a cut-point score of 10 in PHQ 9. We used a cut-off 5 to include mild form depression. However, the majority of literature conducted in Ethiopia used a 5 cut-off point of PHQ 9. So, we add the appropriate reference for this study. Please see page 6

4. Similarly, the citation made for PSS-10 does not provide any of the cut-offs mentioned in the manuscript. You may provide correct reference or re-analyze the data using stress scores as continuous variable.

Ok! We provide the correct reference for PSS-10. Please see page-6

5. It is not clear why the interviews were used rather than self-administered data collection. In addition, what was the setting of data collection? What is the potential effect of using health professionals as interviewers of this sensitive issue? The limitation needs to address any biases arising from this.

Thank you! As you have seen from the socio-demographic part of the result, 25.5% of study participants hadn’t formal education. Due to this reason, we can’t apply self-administered data collection and they were interviewed in a separate secure place. The main reason for using psychiatry nurses as interviewers was to provide appropriate information and psychoeducation for study participants who had comorbid mental illnesses like depression and anxiety. 

6. The ethics statement need to be included in the manuscript submission system rather than just "human participants". In addition, information on what was done for any participant with suicide risk and alcohol dependence is important ethical issue. Please clarify what was done or why no action was taken.

Thank you, dear editor. For alcohol use disorder, we provide information on the availability of modern treatment for their problem and individuals who were ready (especially individuals who reach the dependency stage) to get the treatment center, send them to a treatment and rehabilitation center. But individuals who weren’t ready to get treatment center, we have informed them, they can be treated after they have prepared themselves to get modern treatment for their problem after we gave psychoeducation. But all individuals who were on hazardous/ harmful alcohol use (based on our screening tool) send to hospital for further diagnosis of alcohol dependency. we asked about suicidal behavior when we assess depression, and we send all participants who had suicidal behavior (ideation, attempt, or plan) to hospital for further diagnosis and treatment.

7. In the abstract, results, and discussion, the authors use terms suggestive of causation which cannot be inferred from this cross-sectional survey. Terms such as 'risk factor', 'predictor', 'odds of developing', 'likely to develop', etc. need to be replaced with 'associated with' or other appropriate terms.

 Dear editor, thank you for your comments /suggestions. We have revised based on your comments. 

8. The discussion needs to provide in depth interpretation and implications than simple comparisons of prevalence. You may consider to provide if there are local factors known to increase alcohol use or indicators of complications such as road traffic accident data, etc. 

Thank you for your constructive comment. Modified based on the comment. Please see page-9

9. One or two decimal places are enough for the odds ratios.

 The odds ratio decimal also modified.

2/ For Reviewer 1

Dear reviewer, thank you for the valuable comments and recommendations.

Thank you for your comments. Thank you for your comments on language and we tried to modify the language throughout the paper.

Introduction

1. You put repeated sentence at your introduction and citing different references (reference number 6&7). For example, in recent years, high-income countries have increased their attention to alcohol-related harm by implementing governmental regulations to reduce alcohol consumption. In recent years, high-income countries have increased their attention to alcohol-related harm by implementing governmental regulations to reduce alcohol consumption.

Thank you for the comments and modify based on your comment. Please, see the introduction part. Please see page-3

Methods and materials

1. Better to say "study setting" rather than study settings

Thank you, and we correct it as “Study setting”. Please, see the method part.

Study participants

1. Have you included participants with mental illness?

Yes! Participants who had mental illness and can be communicated and give appropriate information was included in our study. But Participants who were clinically diagnosis for known severe mental illness (who received a clinical diagnosis from mental health professionals or psychiatrists) who cannot give appropriate information due to their psychosis, low energy, motivation/hyperactivity and also difficult to give an interview due to complaining of any discomfort or pain, instability was excluded from the study.

2. Under study setting and populations, it is good to mention the numbers and the type of health institutions in South Gondar Zone. In addition to this I strongly recommend you to include the type of mental health services delivered in those hospitals

 Thank you! We add the type of mental health services given by each hospital. Please, see at the study setting and populations part.

Measurements

1. Why you preferred to measure alcohol used disorder by using CAGE other than AUDIT? More clarification is required.

Thank you for your question. CAGE questionnaire is only assessed individuals who reached at dependency or abuse stage but it does not assessed individuals who had hazardous/harmful alcohol use disorder. The other reason it is not validated in Ethiopian setting, but AUDIT was validated in neighboring African countries. Additionally, the majority of previous Ethiopian studies were carried out based on AUDIT.

Data collection

2. How study participants were reached e.g., door to door knocking/ or telephone interview? 

The participants were interviewed by the nocking door-to-door and they were interviewed in a separate secure place.

Discussion

3. In paragraph 1 and 2 of your discussion were mentions Brazil twice in contradicting sentences " However, the finding of the current study was lower than a study conducted in Brazil 50% (15), and Conversely, the finding of this study was higher than the findings in USA 13.9% (13), India, 9.4% (41), Uganda 9.8% (20), Colombia 9% (42), Brazil, 18.4% (14) ... 

Great thanks! The contradicting information has been modified. Please see the discussion part of page-9

4. On paragraph 4 of your discussion, References are particularly useful to substantiate statements which could be considered strengthen your justification e.g. "The possible reason might be due to lack of experience in social relationship, social and psychological support from their neighborhood and relatives can lead to use alcohol"

 Thank you for constructive comment! You are right, but we believe that if the explanation/justification is represented for the identified problem, a reference is may not be mandatory.

5. How did you manage the participant who are eligible for dependence treatment during the data collection time?

For alcohol use disorder, we provide information on the availability of modern treatment for their problem and individuals who were ready to get treatment center, send them to a treatment and rehabilitation center for further diagnosis and treatment. But individuals who weren’t ready to get treatment center, we have informed them, they can be treated after they have prepared themselves to get modern treatment for their problem.

3/ For reviewer 2

Thank you for your constructive comments.

1. The statement of the problem at the last end of the of the background is not convincing because the community is part of the general population.

Thank you! But if we understand your comment, we attempt to justify the importance of community-based researches to enforce the government than institution-based researches.

2. The authors used sociodemographic data in the analysis, but have not mentioned the instrument used to collect that demographic data under measurements. 

Thank you for your constructive comment! There is no standardized tool to assess sociodemographic characteristics. We adopted from different kinds of literature. So, we included based on your comment. Please see on page 5. 

3. Under results on sociodemographic, the author uses figures to start sentence, they need to write those figures in words.

Thank you! We put the figures and tables separately. So, the interpretation of all figures is available in the main manuscript pages 5 and 7. 

4. Under limitation, the authors failed to indicate the strength of their research and the application of their findings to solving the alcohol disorder in the study district.

Thank you for your constructive comment! Modified based on your comment. Please see page 11.

---

## [Decision Letter · Decision Letter 1]

13 Sep 2021

Magnitude and correlates of alcohol use disorder in south Gondar zone, northwest Ethiopia: a community based cross-sectional study

PONE-D-21-07650R1

Dear Dr. Legas,

We’re pleased to inform you that your manuscript has been judged scientifically suitable for publication and will be formally accepted for publication once it meets all outstanding technical requirements.

Kind regards,

Markos Tesfaye, M.D., Ph.D

Academic Editor

PLOS ONE

Additional Editor Comments (optional):

Thank you for addressing the comments.

Reviewers' comments:

Reviewer's Responses to Questions

**Comments to the Author**

1. If the authors have adequately addressed your comments raised in a previous round of review and you feel that this manuscript is now acceptable for publication, you may indicate that here to bypass the “Comments to the Author” section, enter your conflict of interest statement in the “Confidential to Editor” section, and submit your "Accept" recommendation.

Reviewer #2: All comments have been addressed

2. Is the manuscript technically sound, and do the data support the conclusions?

Reviewer #2: Yes

3. Has the statistical analysis been performed appropriately and rigorously? 

Reviewer #2: Yes

4. Have the authors made all data underlying the findings in their manuscript fully available?

Reviewer #2: No

5. Is the manuscript presented in an intelligible fashion and written in standard English?

Reviewer #2: Yes

6. Review Comments to the Author

Reviewer #2: The manuscript has been revised and is now appropriate to be considered for publication in your journal.

7. PLOS authors have the option to publish the peer review history of their article (what does this mean?). If published, this will include your full peer review and any attached files.

Reviewer #2: No

---

## [Editor Report · Acceptance letter]

22 Sep 2021

PONE-D-21-07650R1 

Magnitude and correlates of alcohol use disorder in south Gondar zone, northwest Ethiopia: a community based cross-sectional study 

Dear Dr. Legas:

I'm pleased to inform you that your manuscript has been deemed suitable for publication in PLOS ONE. Congratulations! Your manuscript is now with our production department. 

Kind regards, 

on behalf of

Prof. Markos Tesfaye 

Academic Editor

PLOS ONE